# Gendered Impact of Age, Toilet Facilities, and Cooking Fuels on the Occurrence of Acute Respiratory Infections in Toddlers in Indonesia and the Philippines

**DOI:** 10.3390/ijerph192114582

**Published:** 2022-11-07

**Authors:** Lilis Sulistyorini, Chung-Yi Li, Leka Lutpiatina, Ricko Dharmadi Utama

**Affiliations:** 1Department of Environmental Health, Faculty of Public Health, Universitas Airlangga, Jalan Mulyosari, Surabaya 60115, Indonesia; 2Department of Public Health, College of Medicine, National Cheng Kung University, Tainan 704, Taiwan; 3Medical Laboratory Technology Poltekkes Kemenkes Banjarmasin, Mistar Cokrokusumo Street 4a, Banjarbaru 70714, Indonesia

**Keywords:** acute respiratory infections, under-five children, Indonesia-Philippines, gender risk factor, Demographic and Health Survey, year 2017

## Abstract

Introduction: The purpose of the study was to analyze the risk factors of acute respiratory infections (ARI) in children under five in Indonesia and the Philippines and to implement a primary review of the characteristics of toddlers and their households in both countries. Methods: Data were obtained from the 2017 Demographic and Health Survey (DHS) of Indonesia and the Philippines. The characteristics of children, mothers, and households were analyzed using bivariate and multivariate logistic regression to examine the significant correlations between variables. Results: The percentage of children under five with ARI symptoms in 2017 was 1.51% in the Philippines and 4.22% in Indonesia. In Indonesia, males aged under one year had significantly lower occurrences of ARI symptoms (OR 0.54, 95% CI 0.36–0.82). In the Philippines, dirty cooking fuels had a significant effect on increasing the likelihood of ARI in female toddlers (OR 4.01, 95% CI 1.02–15.83). In the Philippines, the unavailability of toilet facilities had a significant effect on increasing the likelihood of ARI in male toddlers (OR 2.67, 95% CI 1.15–6.16). Conclusion: The comparison of risk factors between male and female toddlers revealed different results in some variables, as follows: children aged under one year, dirty cooking fuels, and unavailable toilet facilities. The role of parents is fundamental in taking care of female toddlers, since they are more exposed to ARI at the age of under one year and are more prone to indoor air pollution from solid cooking fuels.

## 1. Introduction

Acute respiratory infections (ARI) are diseases that directly affect tissue oxidation, leading to complications that may require quick intervention and intensive care [1]. Among respiratory infections, acute lower respiratory tract infections are a major cause of morbidity, while upper respiratory tract infections are the most common infections, especially in children [2,3]. Most ARI agents (90%) come from viruses [4]. The most frequently detected viruses in children with ARI include respiratory syncytial virus, influenza virus types A and B, parainfluenza virus, adenovirus, human rhinovirus/enterovirus, and human metapneumovirus [4,5,6]. Recently, a severe respiratory distress syndrome known as the novel coronavirus (SARS-CoV-2) has emerged, causing the COVID-19 respiratory infection [7,8].

The main cause of morbidity and mortality worldwide is acute respiratory infections [4]. Pneumonia causes 18% of the annual global mortality rate in children under five [9]. Every year, acute respiratory infections account for 1.9 to 2.2 million childhood deaths and 12 million hospital admissions, 70% occurring in Africa and Southeast Asia [3,10]. The Philippines and Indonesia are 2 of 15 countries that together account for 75% of global pneumonia cases in children [11]. Studies show that the causing agents of ARI vary according to geography and climatic conditions [12]. Therefore, epidemiological data on ARI cases are fundamental in every single region to develop proper prevention and control strategies [13]. Every region has its demographic and socioeconomic characteristics, thus each region has different risk factors for infections in children [14].

The most common risk factors for acute respiratory infections are demographic characteristics, socioeconomic status, environment, and nutrition [15]. Studies in various countries have presented several risk factors for ARI such as population density, gender, place of residence, illiterate mothers, smoking parents, indoor air pollution, cooking fuels, lack of toilet facilities, malnutrition, and family history of ARI [15,16,17,18]. In China, a study focused on the gender impact of solid fuel use and paternal smoking on the occurrence of ARI in children under five revealed that girls had a higher risk of experiencing ARI symptoms when exposed to solid fuels (OR = 3.28; 95% CI 1.34–8.03) or paternal smoking (OR = 2.27; 95% CI 1.08–4.77), while there was no significant effect on boys [19].

Previous research using the 2013 to 2022 Demographic and Health Survey (DHS) in the Philippines presented many risk factors of ARI in toddlers [14]; however, the study did not compare the differences in risk factors between the genders of children under five. Hence, more research is required to examine the possible differences in risk factors for the incidence of ARI between male and female toddlers in two neighboring countries. The purpose of this study was to analyze the risk factors for ARI in children under five in both Indonesia and the Philippines and to conduct a primary review of the characteristics of children and households in both countries.

## 2. Methods and Materials

### 2.1. Data Sources

The research was a cross-sectional study and used the Indonesia Demographic and Health Survey and the Philippines Demographic and Health Survey from 2017.

### 2.2. Methodology

The study used both bivariate and multivariate analysis. The relationship between ARI in children under five and each variable was analyzed by bivariate analysis, while the significant impact of socioeconomic status and the characteristics of children, mothers, and households were analyzed by multivariate logistic regression analysis. Data were presented in an adjusted odds ratio (AOR) with a 95% confidence interval (CI). Stata version 15 was used for data analysis.

### 2.3. Outcome Variables

Data from 10,171 (Filipino data) and 16,623 (Indonesian data) toddlers were obtained from a women’s health questionnaire. Data on the respiratory health of children aged 0 to 59 months were collected by questionnaire. The questionnaire targeted women aged 15 to 49 years. The prevalence of ARI was measured by reporting the child’s condition in the two weeks before the survey by asking, “did he experience any of these symptoms: cough, shortness of breath, rapid breathing, and fever?”

### 2.4. Explanatory Variables

Research variables were the characteristics of children, mothers, and households. The characteristics of children included sex: male, female; age group: under one year, 1–2 years, and (3–4) years; birth order category: 1st–2nd, 3rd–4th, and more than 4th. 

The characteristics of mothers were maternal age group: 15–19 years, 20–24 years, 25–29 years, 30–34 years, 35–39 years, 40–44 years, and 45–49 years; mother’s educational level: not attending school, not completing the primary and secondary educational levels, completing primary and secondary educational levels, and higher education; and the mother’s employment status category: working mothers and non-working mothers. 

The characteristics of households consisted of wealth quintiles: poorest, poorer, middle, richer, richest (the distribution of scores based on the number and types of items owned, ranging from televisions to bicycles or cars, and housing characteristics such as drinking water sources, latrine facilities, and primary building materials for the house floor); area of residence: urban and rural areas; familial smoking behavior in the house; cooking fuels: clean (electricity/gas/kerosene), unclean (coal/charcoal/wood/straw), and not cooking food; drinking water quality: good (plumbing/drinking water company/protected well/bottled water), bad (rivers/lakes/dams/irrigation water/wells are not protected); and handwashing and availability of toilet facilities. 

### 2.5. Research Ethics

Ethical approval was obtained from ICF International. Permission to use and analyze 2017 DHS data was carried out by registering the study on the website of the Demographics and Health Survey (DHS).

## 3. Results

Table 1 shows the distribution and percentage of children with ARI symptoms in the Philippines and Indonesia in 2017. The percentage of children with ARI symptoms in the Philippines (1.51%) was lower than in Indonesia (4.22%). That percentage is almost similar in males and females in both countries; the percentages of boys and girls in the Philippines were 52.36%, and 47.64%, while in Indonesia they were 50.78% and 49.22%, respectively. Maternal characteristics showed that the percentage of mothers with high educational levels in the Philippines (29.5%) was higher than in Indonesia (15.21%). Households’ characteristics revealed that the percentage of the very poor wealth quintile in the Philippines (28.1%) was higher than in Indonesia (20.08%). The percentage of households using dirty cooking fuels in the Philippines (57.02%) was more than in Indonesia (22.85%). Family members who smoke at home in the Philippines (47.88%) were less than in Indonesia (76.26%). The percentage of households without toilet facilities was almost the same in the Philippines (8.5%) and Indonesia (9.62%).

Table 2 shows the effects of the variables on the susceptibility to ARI symptoms in toddlers. Gender had a significant influence in the Philippines, with a *p*-value of 0.0013, and a nonsignificant influence in Indonesia. On the other hand, child age had a significant influence in Indonesia, with a *p*-value of 0.0036, and a nonsignificant influence in the Philippines. The maternal educational level had a significant influence in the Philippines, with a *p*-value of 0.0183, as well as in Indonesia, with a *p*-value of 0.0147. The wealth quintile indicator—the richer the family, the lesser the risk of experiencing ARI symptoms—showed a significant influence in Indonesia, with a *p*-value of 0.0000, but a nonsignificant influence in the Philippines. The cooking fuels indicator—families who cook with 5.08% dirty fuels account for more children with ARI symptoms—had a significant influence, with a *p*-value of 0.0171 in Indonesia, but a nonsignificant influence in the Philippines. The unavailability of toilet facilities had a significant influence in both the Philippines, with a *p*-value of 0.0001, and Indonesia, with a *p*-value of 0.0000. 

Table 3 shows that in both the Philippines and Indonesia, female toddlers were less prone to ARI symptoms; the results were statistically significant in the Philippines (OR: 0.50, 95% CI 0.33–0.77), but not significant in Indonesia (OR: 0.87, 95% CI 0.72–1.05). Children under one year had a lower risk of experiencing ARI symptoms in the Philippines, but the coefficient was not statistically significant (OR: 0.60, 95% CI 0.28–1.27), while it was statistically significant in Indonesia (OR: 0.61, 95% CI 0.45–0.82). Young maternal age (20–24) years significantly increased the risk of children experiencing ARI symptoms in the Philippines (OR: 2.81, 95% CI 1.29–6.07), but insignificantly in Indonesia (OR: 1.04, 95% CI 0.78–1.39). In the Philippines, the mothers who did not graduate from elementary school had a significantly greater probability of having children with ARI symptoms (OR: 2.85, 95% CI 1.29–6.17), while the data were not significant in Indonesia (OR: 1.12, 95% CI 0.73–1.71). Children of non-working mothers had less risk of experiencing ARI symptoms in the Philippines, but the OR is insignificant (OR: 0.83, 95% CI 0.52–1.35), while the OR was significant in Indonesia (OR: 0.80, 95% CI 0.65–0.99). 

The richest wealth quintile was associated with a higher risk of the occurrence of ARI in toddlers in the Philippines (OR: 1.60, 95% CI 0.59–4.36), while the same category significantly decreased the risk of experiencing ARI symptoms in toddlers in Indonesia (OR: 0.61, 95% CI 0.42–0.89). Urban areas had a lower probability of developing ARI symptoms in children under five in the Philippines (OR: 0.92, 95% CI 0.54–1.57), but urban areas had a higher possibility of developing ARI symptoms in children under five in Indonesia (OR: 1.11 95% CI 0.89–1.38). Using dirty cooking fuels significantly increased the likelihood of developing ARI symptoms in toddlers in the Philippines (OR: 1.99, 95% CI 1.11–3.56), while in Indonesia, the data were insignificant (OR: 1.09, 95% CI 0.85–1.40). Unavailable toilet facilities increased ARI symptoms in children under five in the Philippines (OR: 1.89, 95% CI 0.96–3.72) and in Indonesia (OR: 1.28, 95% CI 0.97–1.68).

Table 4 and Table 5 presented a comparison of risk factors between male and female toddlers. In the Philippines, children under one year had less possibility of experiencing ARI symptoms, with OR values that were not different for boys (OR: 0.61, 95% CI 0.25–1.49) and girls (OR: 0.63, 95% CI 0.16–2.51). In Indonesia, boys and girls under one year were less prone to ARI symptoms, with a coefficient that was statistically significant for males (OR: 0.54, 95% CI 0.36–0.82) and not significant for females (OR: 0.72, 95% CI 0.44–1.17).

In the Philippines, the odds ratio of the effects of using dirty cooking fuels on the occurrence of ARI symptoms in female toddlers was higher and statistically significant (OR: 4.01, 95% CI 1.02–15.83), while the odds ratio was lower and insignificant in male toddlers (OR 1.4, 95% CI 0.68–2.93). In Indonesia, using dirty cooking fuels decreased the possibility of experiencing ARI symptoms in toddlers with an odds ratio that was not too different between males (OR 0.89, 95% CI 0.65–1.21) and females (OR 0.91, 95% CI 0.63–1.32). 

In the Philippines, the unavailability of toilet facilities significantly increased the likelihood of the occurrence of ARI symptoms in male toddlers (OR: 2.67, 95% CI 1.15–6.16), while it decreased the likelihood of ARI symptoms in female toddlers (OR: 0.81, 95% CI 0.27–2.38). In Indonesia, the unavailability of toilet facilities elevated the possible occurrence of ARI symptoms with a slightly different odds ratio between male toddlers (OR: 1.37, 95% CI 0.93–2.03) and female toddlers (OR: 1.18, 95% CI 0.76–1.82).

## 4. Discussion

DHS 2017 data showed that ARI in the Philippines (1.51%) was lower than in Indonesia (4.22%). However, previous studies showed that the percentage of ARI in toddlers was 5.7% in 2013 in the Philippines [14] and 5.12% in 2012 in Indonesia [20]. It can be concluded that the number of ARI cases in under-five children is much lower in the Philippines compared to Indonesia.

Both Indonesia and the Philippines are neighboring countries with many islands. However, the population characteristics in both countries are different, as shown in Table 1. The maternal higher educational level was greater in the Philippines than in Indonesia, while the very poor wealth quintile was higher in the Philippines than in Indonesia. The number of households using dirty cooking fuels was higher in the Philippines than in Indonesia. The differences in characteristics in both countries were associated with differences in the impact of the variables on the occurrence of ARI symptoms in children under five, as shown in Table 2. Because of the variation in demographic characteristics and socioeconomic status among regions, each region has its unique risk factors for infections among children [14].

Table 3 showed that female toddlers were less prone to experience ARI symptoms than male toddlers in both the Philippines and Indonesia. This aligns with many studies showing that females under five were less exposed to ARI, such as a study by Sk R et al. (2019) in Afghanistan (OR 0.92 95% CI 0.85–0.99), a study by Dagne H et al. (2020) in Ethiopia (OR 0.98 95% CI 0.64, 1.50), and a study by Mandal S et al. (2020) in India (OR 0.86 95% CI 0.82–0.90) [16,21,22].

Using dirty cooking fuels increased the likelihood of developing ARI symptoms in children under five with a statically significant odds ratio in the Philippines (OR: 1.99, 95% CI 1.11–3.56) but an insignificant odds ratio in Indonesia (OR: 1.09, 95% CI 0.85–1.40). Similarly, a study by Addisu A et al. (2021) in Ethiopia recorded a high odds ratio showing that dirty cooking fuels increased the likelihood of developing ARI symptoms in children (OR 4348, 95% CI 1632, 11,580) [23]. Other studies also presented the same results, such as a study by Mulambya NL et al., (2020) in Zambia (OR of 2.67, 95% CI 2.09–3.42) and a study by Mondal D and Paul P. (2020) in India (OR of 1.10, 95% CI 1.01–1.20) [24,25].

Unavailable toilet facilities had a positive influence on the occurrence of ARI symptoms in children under five in both the Philippines (OR: 1.89, 95% CI 0.96–3.72) and Indonesia (OR: 1.28, 95% CI 0.97–1.68). Similarly, a study by Sk R et al. (2019) in Afghanistan showed that unavailable toilet facilities increased the occurrence of ARI symptoms in children (OR of 1.01, 95% CI 0.90–1.14) [16]. This agrees with other studies showing that the availability of toilet facilities certainly reduced the experience of ARI symptoms in children, such as a study by Hasan Md et al. (2019) in Bangladesh (OR of 0.84, 95% CI 0.74–0.96) and a study by Akinyemi, JO, and Morakinyo OM. (2018) in Nigeria (OR of 0.93, 95% CI 0.77–1.12) [26,27].

The factor of the age of children under one year had a significant effect on decreasing the likelihood of ARI symptoms in boys in Indonesia (Table 5). Males under one year showed a significant decrease in experiencing ARI symptoms, which can be linked to the immune factor during early life. Many studies showed that the immune response was higher in male infants than in female infants. Furthermore, neonatal immunity has been investigated using cord blood samples. The number of monocytes and basophils in male infants was higher than in female infants until the age of 13 months [28], and the frequency of natural killer (NK) cells was greater in male infants than in female infants [29]. Moreover, the level of umbilical cord blood IgE in male neonates was higher than in females [30]. Another study showed that proinflammatory response after stimulation with mitogens or lipopolysaccharide was also greater in male infants than female infants (Casimir, G. J, 2010) [31].

Dirty cooking fuels had a significant effect on increasing the likelihood of ARI symptoms in girls in the Philippines (Table 4). Research by Chen C and Modrek S. (2018) in China also showed a significant result (OR of 3.28, 95% CI 1.34–8.03) [19]. Exposure to solid cooking fuels, indoor air pollution, and paternal smoking increased the potential risk of experiencing fever and coughing among girls [19]. Exposure to cigarette smoke caused respiratory tract irritation, leading to local epithelial damage [32], in addition to increasing the risk of bacterial pathogenesis infection in children [33]. Girls frequently stay indoors with their mothers at cooking time, thus they are more exposed to solid fuel pollution, while boys spend more time outdoors, so their exposure to solid fuel pollution is limited [19].

The unavailable toilet facilities had a significant effect on raising the probability of ARI symptoms in boys in the Philippines (Table 4). People who do not have toilets in their homes use public or shared toilets that are used by several households. The poor sanitation of public toilets enhanced the possibility of pathogen transmission through splashing toilet water and the formation of aerosols [34].

Several respiratory viruses, including influenza virus, coronavirus, and rhinovirus come from feces [35,36], and their infection can be spread by aerosolization during toilet flushing [37]. Aerosols can come from feces and urine, and viruses such as the coronavirus can last a long time in water and sewage [38,39]. The transmission ability of the virus varies according to the number of viruses in feces and urine [40,41,42]. Several studies showed the possibility of the transmission of coronavirus during toilet usage [43,44,45,46]. Respiratory viruses can also be transmitted through feces by fomite transmission, defined by touching objects in the toilet such as doorknobs, dippers, buckets, and faucets [47]. Boys are not patient in spending enough time in the toilet and they are more likely to touch objects in the toilet. However, girls showed toilet-training skills at an earlier age than boys [48].

Public WC facilities or shared WC for several households can increase the possibility of ARI in boys, so support from the government, especially the health office, is needed to provide public toilet facilities that are more appropriate and clean for low-income communities.

The limitation of the study is that the data were cross-sectional, so the causal relationship between factors has not been assessed. Moreover, collecting data through a questionnaire to housewives could be subjective, based on the mother’s perception of the disease without validation from health workers. The possibility of data bias may negatively affect the prevalence of ARI symptoms. 

The strength of the research is that the study explored the comparison of risk factors related to experiencing ARI symptoms among both genders of toddlers in two countries. Therefore, the results of the comparison could be used to assess the advantages and disadvantages of both countries. Moreover, the Demographic Health Survey (DHS) data collection method undergoes a validation process, so the results can be generalized. Similarly, in various countries, the DHS variables are identified to obtain comparable results among countries.

## 5. Conclusions

There were differences in the characteristics of children, mothers, and households of children under five experiencing ARI symptoms in both countries: Indonesia and the Philippines. In Indonesia, the factor of the age of children under one year had a significant effect in reducing the possibility of the occurrence of ARI symptoms in boys. In the Philippines, the factor of dirty cooking fuels had a significant influence on elevating the likelihood of experiencing ARI symptoms in girls. In the Philippines, the factor of unavailable toilet facilities had a significant effect on increasing the probability of the occurrence of ARI symptoms in boys. Parents should pay more attention to female toddlers because they are more vulnerable to ARI at the age of under one year and are more exposed to indoor air pollution from solid cooking fuels. On the other hand, families should pay attention to male toddlers when they are in the toilet. Support from the government, especially the health department, is required to provide clean and proper public toilets to reduce the possibility of pathogen transmissions that cause ARI.

## Figures and Tables

**Table 1 ijerph-19-14582-t001:** Socio-demographic characteristics of participants in the Philippines and Indonesia in 2017.

Variable	Philippines		Indonesia	
	*n*	% ^¥^	*n*	% ^¥^
Child’s Characteristics				
Children with ARI Symptoms				
Yes	159	1.51	744	4.22
No	10,012	98.49	15,879	95.78
Sex				
Male	5306	52.36	8520	50.78
Female	4865	47.64	8103	49.22
Age				
Under 1 year old	1951	19.3	3205	19.12
1–2 years old	3947	38.92	6698	40.44
3–4 years old	4273	41.77	6720	40.44
Child Birth Order				
1st–2nd	5607	57.82	10,635	68.69
3rd–4th	2867	27.86	4686	25.82
More than 4th	1697	14.32	1302	5.5
Mother’s Characteristics				
Age in Years				
15–19	384	3.99	394	2.23
20–24	2137	21.44	2549	16.18
25–29	2810	27.43	4247	25.63
30–34	2177	21.72	4427	26.42
35–39	1644	15.87	3315	19.83
40–44	820	7.9	1395	7.99
45–49	199	1.65	296	1.72
Educational Level				
No education	178	1.12	240	1.08
Incomplete primary	1156	9.49	1158	6.25
Complete primary	934	8.37	2968	19.49
Incomplete secondary	1813	17.08	4283	28.29
Complete secondary	3225	34.43	5021	29.68
Higher	2865	29.5	2953	15.21
Mother’s occupation				
Not working	5487	54.17	7865	49.29
Working	4684	45.83	8758	50.71
Household’s Characteristics				
Wealth Quintile				
Poorest	3726	28.1	4517	20.08
Poorer	2411	22.07	3266	20.17
Middle	1737	19.71	3087	20.46
Richer	1345	16.82	2929	20.18
Richest	952	13.3	2824	19.11
Place of Residence				
Rural	6946	55.63	8425	51.34
Urban	3225	44.37	8198	48.66
Cooking Fuels				
Clean	3103	42.97	11,248	77.05
Unclean	7066	57.02	5355	22.85
Not cooking food	2	0.01	20	0.11
Household Smokers				
Yes	4788	47.88	13,029	76.26
No	5383	53.83	3 594	23.74
Drinking Water Quality				
Good	9339	94.8	9785	63.52
Bad	832	5.2	6838	36.48
Handwashing				
Observed	9492	92.89	15,573	94.47
Not observed	679	7.11	1050	5.53
Toilet facility				
Available	9062	91.5	14,858	90.38
Not available	1109	8.5	1765	9.62
*n* Total	10,171		16,623	

**^¥^** Weighted proportion; source: the Philippines and Indonesia DHS 2017.

**Table 2 ijerph-19-14582-t002:** Relationship between the characteristics of children, mothers, and households in the Philippines and Indonesia in 2017.

Characteristic	Philippines					Indonesia				
	Children with ARI Symptoms		Children without ARI Symptoms		*p*-Value	Children with ARI Symptoms		Children without ARI Symptoms		*p*-Value
	n	% ^¥^	n	% ^¥^		n	% ^¥^	n	% ^¥^	
Child’s Characteristics										
Sex					0.0013 *					0.1854
Male	97	1.96	5209	98.04		397	4.46	8123	95.54	
Female	62	1.02	4803	98.98		347	3.96	7756	96.04	
Age					0.2303					0.0036 *
Under 1 year old	25	1.0	1926	99.0		105	3.12	3100	96.88	
1–2 years old	70	1.8	3877	98.2		349	4.89	6349	95.11	
3–4 years old	64	1.48	4209	98.52		290	4.06	6430	95.94	
Child Birth Order					0.2585					0.7087
1st–2nd	82	1.39	5525	98.61		473	4.17	10,302	95.83	
3rd–4th	47	1.39	2820	98.61		205	4.21	4481	95.79	
More than 4th	30	2.24	1667	97.76		66	4.85	1236	95.15	
Mother’s Characteristics										
Age in Years					0.0658					0.2761
15–19	8	1.34	376	98.66		17	4.47	377	95.53	
20–24	38	2.05	2099	97.95		136	4.68	2413	95.32	
25–29	43	1.18	2767	98.82		200	4.34	4047	95.66	
30–34	28	0.77	2149	99.23		201	4.59	4226	95.41	
35–39	26	1.8	1618	98.2		124	3.46	3191	96.54	
40–44	15	2.76	805	97.24		58	3.95	1337	96.05	
45–49	1	1.59	198	98.41		8	2.02	288	97.98	
Educational Level					0.0183 *					0.0147 *
No education	0	0	178	100.0		15	5.7	225	94.3	
Incomplete primary	38	3.13	1118	96.87		62	5.67	1096	94.33	
Complete primary	15	2.16	919	97.84		162	5.11	2806	94.89	
Incomplete secondary	25	1.07	1788	98.93		201	4.3	4082	95.7	
Complete secondary	46	1.6	3179	98.4		183	3.44	4838	96.56	
Higher	35	1.02	2830	98.98		121	3.73	2832	96.27	
Mother’s Occupation					0.4370					0.106
Not working	74	1.39	5413	98.61		318	3.88	7547	96.12	
Working	85	1.66	4599	98.34		426	4.54	8332	95.46	
Household’s Characteristics										
Wealth Quintile					0.7274					0.0000 *
Poorest	67	1.72	3659	98.28		279	6.21	4238	93.79	
Poorer	34	1.58	2377	98.42		156	4.94	3110	95.06	
Middle	26	1.59	1711	98.41		113	3.39	2974	96.61	
Richer	18	1.46	1327	98.54		107	3.51	2822	96.49	
Richest	14	0.92	938	99.08		89	2.99	2735	97.01	
Place of Residence					0.3655					0.0619
Rural	113	1.66	6833	98.34		428	4.6	7997	95.4	
Urban	46	1.32	3179	98.68		316	3.81	7882	96.19	
Cooking Fuels					0.0695					0.0171 *
Clean	37	1.1	3066	98.9		469	3.97	10,919	96.03	
Unclean	122	1.83	6944	98.17		274	5.08	5081	94.92	
Not cooking food	0	0	2	100.0		1	1.1	19	98.9	
Household smokers					0.7817					0.081
Yes	78	1.57	4710	98.43		625	4.52	12,544	95.48	
No	81	1.47	5302	98.53		119	3.26	3475	96.74	
Drinking Water Quality					0.5950					0.2030
Good	148	1.49	9191	98.51		406	4.01	9379	95.99	
Bad	11	1.83	821	98.17		338	4.57	6500	95.43	
Handwashing					0.0247 *					0.3157
Observed	148	1.52	9344	98.48		697	4.17	15,016	95.83	
Not observed	11	1.46	668	98.54		47	5.05	1003	94.95	
Toilet Facility					0.0001 *					0.0000 *
Available	133	1.39	8929	98.61		628	3.95	14,370	96.05	
Not available	26	2.83	1083	97.17		116	6.73	1649	93.27	

* *p*-value < 0.05; **^¥^** weighted proportion; source: the Philippines and Indonesia DHS 2017.

**Table 3 ijerph-19-14582-t003:** Prediction of children with ARI symptoms in the Philippines and Indonesia in 2017.

Variable	Philippines			Indonesia		
	OR	95% CI		OR	95% CI	
		Lower	Upper		Lower	Upper
Child’s Characteristics						
Sex						
Male	1.00			1.00		
Female	0.50 ***	0.33	0.77	0.87	0.72	1.05
Age						
Under 1 year old	0.60	0.28	1.27	0.61 ***	0.45	0.82
(1–2) years old	1.00			1.00		
(3–4) years old	0.80	0.46	1.40	0.82	0.66	1.01
Child Birth Order						
1st–2nd	1.00			1.00		
3rd–4th	1.11	0.57	2.16	1.12	0.86	1.47
More than 4th	1.34	0.46	3.90	1.29	0.86	1.94
Mother’s Characteristics						
Age In Years						
15–19	1.94	0.66	5.73	0.98	0.48	1.97
20–24	2.81 **	1.29	6.07	1.04	0.78	1.39
25–29	1.54	0.79	2.97	0.97	0.74	1.27
30–34	1.00			1.00		
35–39	1.97	0.92	4.23	0.69 *	0.51	0.95
40–44	2.80 *	1.18	6.72	0.76	0.51	1.14
45–49	1.63	0.18	14.42	0.35 *	0.15	0.84
Educational Level						
No education	N/A	N/A	N/A	1.11	0.56	2.20
Incomplete primary	2.85 **	1.29	6.17	1.12	0.73	1.71
Complete primary	1.92	0.79	4.69	1.10	0.82	1.48
Incomplete secondary	1.00			1.00		
Complete secondary	1.56	0.79	3.09	0.92	0.71	1.19
Higher	0.96	0.47	1.98	1.08	0.77	1.51
Mother’s Occupation						
Not working	0.83	0.52	1.35	0.80 *	0.65	0.99
Working	1.00			1.00		
Household’s Characteristics						
Wealth Quintile						
Poorest	0.76	0.41	1.43	1.26	0.94	1.68
Poorer	1.00			1.00		
Middle	1.59	0.83	3.05	0.69 *	0.49	0.95
Richer	2.07	0.84	5.12	0.71	0.51	1.01
Richest	1.60	0.59	4.36	0.61 **	0.42	0.89
Place of Residence						
Rural	1.00			1.00		
Urban	0.92	0.54	1.57	1.11	0.89	1.38
Cooking Fuels						
Clean	1.00			1.00		
Unclean	1.99 *	1.11	3.56	1.09	0.85	1.40
Not cooking food	N/A	N/A	N/A	0.28	0.03	2.33
Household smokers						
Yes	0.92	0.57	1.49	1.23	0.95	1.59
No	1.00			1.00		
Drinking Water Quality						
Good	1.00			1.00		
Bad	0.88	0.41	1.89	1.06	0.85	1.32
Handwashing						
Observed	1.00			1.00		
Not observed	0.82	0.33	2.01	0.89	0.59	1.32
Toilet Facility						
Available	1.00			1.00		
Not available	1.89	0.96	3.72	1.28	0.97	1.68

* *p*-value < 0.05; ** *p*-value < 0.01; *** *p*-value < 0.001; source: the Philippines and Indonesia DHS 2017.

**Table 4 ijerph-19-14582-t004:** Predictions of children with ARI symptoms in the Philippines compared by gender in 2017.

Variables	Total			Male			Female		
	OR	95% CI		OR	95% CI		OR	95% CI	
		Lower	Upper		Lower	Upper		Lower	Upper
Child’s Characteristics									
Age									
Under one year old	0.60	0.28	1.27	0.61	0.25	1.49	0.63	0.16	2.51
1–2 years old	1.00			1.00			1.00		
3–4 years old	0.80	0.46	1.40	0.83	0.40	1.72	0.78	0.40	1.52
Child Birth Order									
1st–2nd	1.00			1.00			1.00		
3rd–4th	1.11	0.57	2.16	0.99	0.45	2.20	1.29	0.38	4.39
More than 4th	1.34	0.46	3.90	1.42	0.39	5.09	1.23	0.21	7.21
Mother’s Characteristics									
Age in Years									
15–19	1.94	0.66	5.73	2.74	0.71	10.52	1.02	0.20	5.15
20–24	2.81 **	1.29	6.07	3.48 **	1.39	8.74	1.86	0.48	7.23
25–29	1.54	0.79	2.97	1.66	0.66	4.17	1.39	0.51	3.83
30–34	1.00			1.00			1.00		
35–39	1.97	0.92	4.23	2.49	0.99	6.31	1.21	0.30	4.87
40–44	2.80 *	1.18	6.72	3.44 *	1.09	10.87	1.86	0.37	9.46
45–49	1.63	0.18	14.42	3.87	0.45	33.56	N/A	N/A	N/A
Educational Level									
No education	N/A	N/A	N/A	N/A	N/A	N/A	N/A	N/A	N/A
Incomplete primary	2.85 **	1.29	6.17	3.58 *	1.29	9.95	2.05	0.67	6.29
Complete primary	1.92	0.79	4.69	2.65	0.89	7.85	1.19	0.27	5.24
Incomplete secondary	1.00			1.00			1.00		
Complete secondary	1.56	0.79	3.09	1.70	0.67	4.31	1.37	0.51	3.72
Higher	0.96	0.47	1.98	1.32	0.51	3.43	0.45	0.12	1.71
Mother’s Occupation									
Not working	0.83	0.52	1.35	0.99	0.54	1.84	0.61	0.28	1.31
Working	1.00			1.00			1.00		
Household’s Characteristics									
Wealth Quintile									
Poorest	0.76	0.41	1.43	0.79	0.35	1.81	0.72	0.29	1.77
Poorer	1.00			1.00			1.00		
Middle	1.59	0.83	3.05	1.45	0.64	3.29	1.69	0.62	4.59
Richer	2.07	0.84	5.12	2.17	0.77	6.14	1.61	0.28	9.14
Richest	1.60	0.59	4.36	1.29	0.41	4.07	2.91	0.45	18.94
Place of Residence									
Rural	1.00			1.00			1.00		
Urban	0.92	0.54	1.57	0.92	0.46	1.84	0.93	0.43	1.99
Cooking Fuels									
Clean	1.00			1.00			1.00		
Unclean	1.99 *	1.11	3.56	1.4	0.68	2.93	4.01 *	1.02	15.83
No cooking food	N/A	N/A	N/A	N/A	N/A	N/A	N/A	N/A	N/A
Household Smokers									
Yes	0.92	0.57	1.49	0.96	0.52	1.77	0.88	0.42	1.85
No	1.00			1.00			1.00		
Drinking Water Quality									
Good	1.00			1.00			1.00		
Bad	0.88	0.41	1.89	1.04	0.42	2.59	0.72	0.21	2.41
Handwashing									
Observed	1.00			1.00			1.00		
Not observed	0.82	0.33	2.01	0.63	0.21	1.89	1.14	0.26	5.05
Toilet Facility									
Available	1.00			1.00			1.00		
Not available	1.89	0.96	3.72	2.67 *	1.15	6.16	0.81	0.27	2.38

* *p*-value < 0.05; ** *p*-value < 0.01.

**Table 5 ijerph-19-14582-t005:** Predictions of children with ARI symptoms in Indonesia compared by gender in 2017.

Variable	Total			Male			Female		
	OR	95% CI		OR	95% CI		OR	95% CI	
		Lower	Upper		Lower	Upper		Lower	Upper
Child’s Characteristics									
Age									
Under one year old	0.61 ***	0.45	0.82	0.54 **	0.36	0.82	0.72	0.44	1.17
1–2 years old	1.00			1.00			1.00		
3–4 years old	0.82	0.66	1.01	0.83	0.62	1.12	0.80	0.59	1.09
Child Birth Order									
1st–2nd	1.00			1.00			1.00		
3rd–4th	1.12	0.86	1.47	0.97	0.69	1.37	1.27	0.85	1.89
More than 4th	1.29	0.86	1.94	1.39	0.78	2.49	1.09	0.64	1.89
Mother Characteristics									
Age In Years									
15–19	0.98	0.48	1.97	1.76	0.78	3.97	0.21	0.06	0.72
20–24	1.04	0.78	1.39	1.16	0.76	1.77	0.91	0.59	1.40
25–29	0.97	0.74	1.27	1.15	0.80	1.64	0.79	0.53	1.18
30–34	1.00			1.00			1.00		
35–39	0.69 *	0.51	0.95	0.68	0.44	1.05	0.71	0.45	1.10
40–44	0.76	0.51	1.14	0.79	0.48	1.31	0.75	0.41	1.39
45–49	0.35 *	0.15	0.84	0.43	0.14	1.36	0.27	0.07	1.12
Educational Level									
No education	1.11	0.56	2.20	0.51	0.18	1.44	2.38	0.96	5.91
Incomplete primary	1.12	0.73	1.71	0.79	0.43	1.41	1.52	0.84	2.74
Complete primary	1.10	0.82	1.48	1.16	0.77	1.77	1.05	0.70	1.57
Incomplete secondary	1.00			1.00			1.00		
Complete secondary	0.92	0.71	1.19	0.86	0.60	1.23	0.97	0.67	1.42
Higher	1.08	0.77	1.51	0.94	0.59	1.48	1.25	0.76	2.06
Mother’s Occupation									
Not working	0.80 *	0.65	0.99	0.84	0.64	1.11	0.76	0.56	1.04
Working	1.00			1.00			1.00		
Household’s Characteristics									
Wealth Quintile									
Poorest	1.26	0.94	1.68	1.30	0.89	1.89	1.18	0.76	1.85
Poorer	1.00			1.00			1.00		
Middle	0.69 *	0.49	0.95	0.69	0.45	1.09	0.66	0.42	1.04
Richer	0.71	0.51	1.01	0.75	0.48	1.17	0.66	0.39	1.11
Richest	0.61 **	0.42	0.89	0.75	0.45	1.25	0.47 **	0.27	0.83
Place of Residence									
Rural	1.00			1.00			1.00		
Urban	1.11	0.89	1.38	1.02	0.79	1.32	1.24	0.88	1.75
Cooking Fuels									
Clean	1.00			1.00			1.00		
Unclean	1.09	0.85	1.40	0.89	0.65	1.21	0.91	0.63	1.32
No cooking food	0.28	0.03	2.33	0.56	0.07	4.67	N/A	N/A	N/A
Household Smokers									
Yes	1.23	0.95	1.59	1.32	0.93	1.87	1.18	0.79	1.76
No	1.00			1.00			1.00		
Drinking Water Quality									
Good	1.00			1.00			1.00		
Bad	1.06	0.85	1.32	1.12	0.84	1.49	0.99	0.73	1.36
Handwashing									
Observed	1.00			1.00			1.00		
Not observed	0.89	0.59	1.32	0.69	0.39	1.23	1.13	0.65	1.96
Toilet Facility									
Available	1.00			1.00			1.00		
Not available	1.28	0.97	1.68	1.37	0.93	2.03	1.18	0.76	1.82

* *p*-value < 0.05; ** *p*-value < 0.01; *** *p*-value < 0.001.

## Data Availability

Data is freely and publicly available at: https://dhsprogram.com/ (accessed on 5 August 2022).

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
