# Peer review of "Gendered Impact of Age, Toilet Facilities, and Cooking Fuels on the Occurrence of Acute Respiratory Infections in Toddlers in Indonesia and the Philippines"

_ijerph, 2022, doi:10.3390/ijerph192114582_

Round 1

Reviewer 1 Report

The article is a bit interesting; however the methodology of women's health questionnaire is questionable.

I have two major concerns with the article that need proper explanation:

1. page 2 of 14 (lines 82-87): How can you confirm that the symptoms of 'cough, shortness of beath, rapid breathing, and fever are directly associated with Acute Respiratory Infections (ARI). Mothers of toddlers may provide wrong information on the association between symptoms and ARI.

2.  page 8 of 14 (lines 213-214): The association of toilet facilities and ARI symptoms seem to be not explained properly. How can you draw the conclusion between facility use and ARI symptoms? please provide more details.

minor: I'd suggest that you add the recent published article by Harvard on CoV-2 on page 1 line 42:

Stern, R., Al-Hemoud, A., Alahmad, B., Koutrakis, P. Levels and particle size distribution of airborne SARS-CoV-2 at a healthcare facility in Kuwait. Science of The Total Environment. 2021; 782, 146799 

Author Response

Response to Reviewer 1 Comments

Point 1: page 2 of 14 (lines 82-87): How can you confirm that the symptoms of 'cough, shortness of beath, rapid breathing, and fever are directly associated with Acute Respiratory Infections (ARI). Mothers of toddlers may provide wrong information on the association between symptoms and ARI.

Response 1: we recognize this as a limitation of the study, and we have included it in the article: page 12 of 15 (lines 280-284) (in red).

Point 2: page 8 of 14 (lines 213-214): The association of toilet facilities and ARI symptoms seem to be not explained properly. How can you draw the conclusion between facility use and ARI symptoms? please provide more details.

Response 2: explanation of public toilet facilities related to the transmission of ARI is listed in the discussion on pages 11 of 15 and 12 of 15 (lines 261-273) (in red).

Point 3: minor: I'd suggest that you add the recent published article by Harvard on CoV-2 on page 1 line 42:

Stern, R., Al-Hemoud, A., Alahmad, B., Koutrakis, P. Levels and particle size distribution of airborne SARS-CoV-2 at a healthcare facility in Kuwait. Science of The Total Environment. 2021; 782, 146799

Response 3: literature we have added to the article on page 1 of 15 (lines 42) and page 13 of 15 (lines 331-333) (in red).

Reviewer 2 Report

Several paragraphs has to be reformulated .They are not very clearly expressed.

Ex: row 38 has to be "include" instead include a"

row 58 "gender " instead "gendered"

row 64 "risk factor between genders " and not " both genders"

row 82 "of " instead "about"

It has to be a more accurate description of the questionnaire used for the study

Is not understandable the definition of " dirty fuel", "poor/poorer/middle/richer/richest " how was defined , drinking water quality definition

row 90 reformulate sex and age group

A huge mistake which has to be revised : at discussion section they are results as well.They are 2 tables presented ( 4 and 5) at discussion .

The discussion paragraph start with a conclusion.Also this has to be corrected.

references : 13,16,19,22,28,32,33,37, 46 are not properly written.Has to be corrected

English also need improvement.

Author Response

Response to Reviewer 2 Comments

Point 1: row 38 has to be "include" instead include a"

Response 1: fixed on page 1 of 15 (line 38) (in red)

Point 2: row 58 "gender " instead "gendered"

Response 2: fixed on page 2 of 15 (line 58) (in red)

Point 3: row 64 "risk factor between genders " and not " both genders"

Response 3: fixed on page 2 of 15 (line 64) (in red)

Point 4: row 82 "of " instead "about"

Response 4: fixed on page 2 of 15 (line 82) (in red)

Point 5: Is not understandable the definition of " dirty fuel", "poor/poorer/middle/richer/richest" how was defined , drinking water quality definition

Response 5: fixed on page 3 of 15 (line 98-106) (in red)

Point 6: row 90 reformulate sex and age group

Response 6: fixed on page 2 of 15 (line 90-91) (in red)

Point 7: A huge mistake which has to be revised : at discussion section they are results as well.They are 2 tables presented ( 4 and 5) at discussion .

Response 7: the table has been moved to the research results section on page 8-10 of 15 (line 189-192) (in red)

Point 8: The discussion paragraph start with a conclusion.Also this has to be corrected.

Response 8: The discussion paragraph does not start with a conclusion, this sentence is a comparison of the results of ARI under five in Indonesia with the Philippines (this is not the conclusion of the article), it is on page 10 of 15 (line 194-196) (in red)

Point 9: references : 13,16,19,22,28,32,33,37, 46 are not properly written.Has to be corrected

Response 9: fixed on page 13-15 of 15 (line 344-428) (in red)

Reviewer 3 Report

The manuscript “Gendered Impact of Age, Toilet Facilities, and Cooking Fuels on the Occurrence of Acute Respiratory Infections in Toddlers in Indonesia and the Philippines” has well written. The study is significant in the context of social determinants of children health. I have minor suggestions.

Abstract

Abbreviate ARI

Methods

Provide a detailed description of the Demographic and Health Survey (periodicity of survey, total sample, total number of children, key variables).

Discussion

The discussion gender dimension of ARI needs to be discussed more.  

Mention one paragraph on Implications for policy and practice.

Author Response

Point 1: Abstract: Abbreviate ARI

Response 1: fixed on page 1 of 15 (line 13-14) (in red)

Point 2: Methods: Provide a detailed description of the Demographic and Health Survey (periodicity of survey, total sample, total number of children, key variables).

Response 2: fixed on page 2-3 of 15 (line 72-106) (in red)

Point 3: Discussion: The discussion gender dimension of ARI needs to be discussed more.

Response 3: fixed on page 11 of 15 (line 241-250) (in red)

Point 4: Discussion: Mention one paragraph on Implications for policy and practice.

Response 4: fixed on page 12 of 15 (line 276-279) (in red)

Round 2

Reviewer 1 Report

The authors have incorporated all suggestions and answered all questions correctly.

Author Response

Thank you for accepting our article improvement

Reviewer 2 Report

Some changes still has to be done.

Row 92 - reformulate more clear that were studied mothers by age groups

Row128 -reformulate " genders influence on ARI"

Row 132 and some others- I recommend replace of" insignificant" with other formulation.

Row 187"not too different" is need to clarify weather it is or not different

In the discussion paragraph often authors talk about results. This has to be revised and not present results which are already presented in results paragraph.

Row 232 - authors present again results - data of table 4, 5 this has to be reconsidered, reformulated

Author Response

Response to Reviewer 2 Comments

Point 1: Row 92 - reformulate more clear that were studied mothers by age groups

Response 1: fixed on page 2 of 14 (line 92) (in red)

Point 2: Row128 -reformulate " genders influence on ARI"

Response 2: fixed on page 4 of 14 (line 128) (in red)

Point 3: Row 132 and some others- I recommend replace of" insignificant" with other formulation.

Response 3: fixed on page 4-5 of 14 (line 129-139) (in red)

Point 4: Row 187"not too different" is need to clarify weather it is or not different

Response 4: fixed on page 8 of 14 (line 187) (in red)

Point 5: Row 232 - authors present again results - data of table 4, 5 this has to be reconsidered, reformulated

Response 5: the results of the table data have been omitted in the discussion paragraph
